Cell kinetics during regeneration in the sponge Halisarca caerulea: how local is the response to tissue damage?

Alexander Brittany E. 1 2 B.E.Alexander@uva.nl britt.e.alexander@gmail.com
Achlatis Michelle 1
Osinga Ronald 2
van der Geest Harm G. 1
Cleutjens Jack P.M. 3
Schutte Bert 4
de Goeij Jasper M. 1 2
1 Department of Aquatic Ecology and Ecotoxicology, Institute for Biodiversity and Ecosystem Dynamics, University of Amsterdam , Amsterdam , The Netherlands
2 Porifarma B.V. , Ede , The Netherlands
3 Department of Pathology, Cardiovascular Research Institute Maastricht, Maastricht University , Maastricht , The Netherlands
4 Department of Molecular Cell Biology, Research Institute Growth and Development, Maastricht University , Maastricht , The Netherlands
Kim Cheorl-Ho
Electronic publication date: 2015 Mar 10
Publication date: 2015
Volume: 3
Electronic Location ID: e820
Received 2014 Dec 23; Accepted 2015 Feb 16
Copyright: © 2015 Alexander et al.
Copyright year: 2015
Copyright holder: Alexander et al.
License: This is an open access article distributed under the terms of the Creative Commons Attribution License, which permits unrestricted use, distribution, reproduction and adaptation in any medium and for any purpose provided that it is properly attributed. For attribution, the original author(s), title, publication source (PeerJ) and either DOI or URL of the article must be cited.
License URL: https://creativecommons.org/licenses/by/4.0/

Keywords: Sponges, Regeneration, Cell kinetics, Trade-off, Immunohistochemistry, Choanocyte turnover, Collagen, Modular integration

Funding: European Union Seventh Framework Programme (FP7/2007–2013) KBBE-2010–266033 Innovational Research Incentives Scheme of the Netherlands Organization for Scientific Research NWO-VENI 863.10.009 Porifarma B.V. received funding from the European Union Seventh Framework Programme (FP7/2007–2013) under grant agreement no. KBBE-2010–266033 to undertake the research leading to these results. Funding was also received from The Innovational Research Incentives Scheme of the Netherlands Organization for Scientific Research (NWO-VENI; 863.10.009; personal grant to Jasper M de Goeij). The funders had no role in study design, data collection and analysis, decision to publish, or preparation of the manuscript.

==============================
Sponges have a remarkable capacity to rapidly regenerate in response to wound infliction. In addition, sponges rapidly renew their filter systems (choanocytes) to maintain a healthy population of cells. This study describes the cell kinetics of choanocytes in the encrusting reef sponge Halisarca caerulea during early regeneration (0–8 h) following experimental wound infliction. Subsequently, we investigated the spatial relationship between regeneration and cell proliferation over a six-day period directly adjacent to the wound, 1 cm, and 3 cm from the wound. Cell proliferation was determined by the incorporation of 5-bromo-2′-deoxyuridine (BrdU). We demonstrate that during early regeneration, the growth fraction of the choanocytes (i.e., the percentage of proliferative cells) adjacent to the wound is reduced (7.0 ± 2.5%) compared to steady-state, undamaged tissue (46.6 ± 2.6%), while the length of the cell cycle remained short (5.6 ± 3.4 h). The percentage of proliferative choanocytes increased over time in all areas and after six days of regeneration choanocyte proliferation rates were comparable to steady-state tissue. Tissue areas farther from the wound had higher rates of choanocyte proliferation than areas closer to the wound, indicating that more resources are demanded from tissue in the immediate vicinity of the wound. There was no difference in the number of proliferative mesohyl cells in regenerative sponges compared to steady-state sponges. Our data suggest that the production of collagen-rich wound tissue is a key process in tissue regeneration for H. caerulea, and helps to rapidly occupy the bare substratum exposed by the wound. Regeneration and choanocyte renewal are competing and negatively correlated life-history traits, both essential to the survival of sponges. The efficient allocation of limited resources to these life-history traits has enabled the ecological success and diversification of sponges.

Introduction

Sponges (Porifera) are highly successful organisms. More than 8,500 species have been described to date (Van Soest et al., 2015) and they are among the oldest extant metazoans on Earth (Müller, 1998; Philippe et al., 2009; Antcliffe, Callow & Brasier, 2014). Sponges often dominate the benthic cover of diverse marine and freshwater ecosystems from tropical to polar regions (e.g., Gili & Coma, 1998; Bell & Barnes, 2000; McClintock et al., 2005). Their high cellular (Müller, 2006; Funayama, 2013) and morphological (Gaino & Burlando, 1990) plasticity allows them to constantly adjust the shape and size of their tissues and adapt to variations in environmental conditions (e.g., Gaino, Manconi & Pronzato, 1995; Bell, Barnes & Turner, 2002). The ecological success of sponges is partially a result of their rapid regeneration capacity enabling them to recover from damage caused by predation (Ayling, 1983), storm (Wulff, 2006a; Wulff, 2006b; Wulff, 2010), and environmental stress (Luter, Whalen & Webster, 2012). Additionally, sponges play an important ecological role in the recycling of energy and nutrients within their ecosystems by rapidly renewing their cells through proliferation and shedding (De Goeij et al., 2009; De Goeij et al., 2013; Alexander et al., 2014).

Sponges pump vast amounts of water through their bodies and their choanocytes (filter cells) are constantly renewed, presumably to prevent damage by exposure to physical, chemical, and biological stress from their environment (De Goeij et al., 2009). The choanocytes of the tropical reef sponge Halisarca caerulea (Porifera: Demospongiae) proliferate rapidly under steady-state, non-growing conditions, with a cell-cycle duration of approximately 6 h (De Goeij et al., 2009). Rapid cell proliferation is balanced with massive amounts of cell shedding into the lumen of the excurrent canals in order to maintain tissue homeostasis in the choanocyte compartment. However, it is unknown how cell renewal in sponges changes in response to external cues. In highly proliferative tissues of other organisms, cell proliferation is often altered in response to starvation, tissue damage, and infection. For example, cell proliferation becomes reduced in response to starvation in the mammalian gastrointestinal tract (Aldewachi et al., 1975; Chaudhary et al., 2000) and the Drosophila melanogaster midgut (McLeod et al., 2010), and increases as a result of tissue damage and infection in both systems (gastrointestinal tract: Gilbert et al., 2015; Boshuizen et al., 2003, D. melanogaster midgut: Buchon et al., 2009; Jiang et al., 2011).

Sponges are considered to exhibit the highest regenerative capacity among metazoans, displaying up to 2,900 times their normal growth rate following injury (Ayling, 1983). Regeneration is an energetically demanding process, demonstrated by an increased metabolic rate after tissue damage in the sponge Haliclona oculata (Koopmans et al., 2011). Energy availability is limited and must be distributed between several potentially competing life-history traits such as maintenance, repair, growth, and reproduction (Zera & Harshman, 2001). Energetic resources are often redirected into regenerative processes after tissue damage, potentially limiting energy available for other life history processes (Henry & Hart, 2005). Somatic growth, sexual reproduction, and chemical defense mechanisms can become compromised during tissue regeneration in sponges (Henry & Hart, 2005; Walters & Pawlik, 2005; Leong & Pawlik, 2010). The relationship between regeneration and rapid cell turnover in sponges, both essential life-history traits, is unknown.

Sponges are modular organisms consisting of multiple repeated units, known as aquiferous modules (Ereskovskii, 2003), which may respond differently to tissue damage depending on their distance from the wound. An aquiferous module is a volume of sponge that is associated with a single osculum and supplied by an anatomically-defined aquiferous system of canals and choanocyte chambers (Fry, 1970; Fry, 1979). The degree of integration between aquiferous modules depends on the number and distribution pattern of oscula, with mono-oscular sponges being the most integrated and multi-oscular sponges being the least integrated (Ereskovskii, 2003). Therefore, in thin, encrusting, multi-oscular sponges such as H. caerulea, tissue damage may have a larger impact on the physiology of aquiferous modules closer to the site of injury.

This study aims to gain insight into the potential trade-off between regeneration and cell proliferation in our model species, H. caerulea. We investigate the impact of tissue damage on choanocyte cell kinetics during initial stages of regeneration and choanocyte proliferation rates at later stages of regeneration along a spatial gradient from the wound. Additionally, we examine the role of mesohyl cell proliferation and collagen production during regeneration.

Materials and Methods

Sponge collection

Specimens of H. caerulea were collected at water depths between 15 and 30 m by SCUBA diving on the reefs of the Caribbean island of Curaçao (12°12′N, 68°56′W) between February and April 2013. Fieldwork was performed under the research permit (#2012/48584) issued by the Curaçaoan Ministry of Health, Environment and Nature (GMN) to the CARMABI foundation. Pieces of sponge were chiseled from the reef framework and the attached coral rock substrate was cleared of other organisms. All sponges were trimmed to a size of approximately 25 cm2. Specimens were kept in 100-L aquaria filled with unfiltered running seawater pumped from 10 m water depth from the reef slope, with a flow rate of 3 L min−1. Aquarium water was at ambient temperature (26–27 °C). Aquaria were kept under natural light cycles and were shaded with semi-transparent black plastic sheets to imitate cryptic light conditions. Sponges were allowed to acclimatize for a minimum of two weeks prior to experimentation. The mean (±SE) distance between oscula was measured in ImageJ from photographs of the sponges in order to estimate if tissue samples were likely to be part of the same aquiferous module.

Induction of tissue damage, BrdU-labeling, tissue fixation and embedding

Tissue damage was induced by removing a small piece of tissue (∼1 cm2) from the center of the sponge using a scalpel, and exposing the bare substrate (Fig. 1A). Wounded sponges were placed in individual incubation chambers (3L) containing magnetic stirring devices. The incubation chambers were kept in the aquaria to maintain ambient seawater temperature (De Goeij et al., 2009; Alexander et al., 2014). In order to determine cell cycle parameters (e.g., length of the cell cycle, growth fraction) of choanocytes directly after wound infliction (herein referred to as ‘early regenerative tissue’), sponges were continuously labeled with 50 µmol L−1 5-bromo-2′-deoxyuridine (BrdU, Sigma) directly after wounding for t = 0, 0.5, 1.5, 2, 6, and 8 h (n = 3 for each time point). After BrdU-labeling, a tissue sample (∼0.5 cm2) directly adjacent to the wound was removed from each sponge (Fig. 1A).

Figure 1 Regeneration in H. caerulea (A) directly, (B) one day, (C) two days, and (D) six days after wound infliction.

After six days, sponges had completely filled in the bare substrate exposed by the initial wound with a thin layer of regenerative wound tissue. Tissue samples were taken directly adjacent to the wound, 1 cm from the wound, and 3 cm from the wound (A). Tissue samples taken adjacent to the wound were marked with an arrow shape so that the orientation of the tissue could be recognized and histological sections could be made that included the wound area (A). Within each tissue sample, three histological sections were analyzed, each 100 µm apart, represented by the solid black lines (A). Photographs by Brittany Alexander.

To determine cell proliferation during regeneration, sponges were left to regenerate for t = 0, 1, 2 and 6 days after wound infliction (n = 3 for each time point), then were continuously labeled with BrdU for 6 h. After BrdU-labeling, three tissue samples (∼0.5 cm2) were removed from each sponge; (1) directly adjacent to the wound, (2) approximately 1 cm from the wound, and (3) approximately 3 cm from the wound (Fig. 1A). Tissue samples adjacent to the wound were marked with an arrow shape so the orientation of the tissue could be recognized and histological sections could be made that included the wound area (Fig. 1A). Tissue was fixed in 4% paraformaldehyde in phosphate-buffered saline (PFA/PBS; pH 7.4–7.6, 4 h at 4 °C), rinsed in PBS, dehydrated through a graded series of ethanol and stored in 70% ethanol at 4 °C until further processing. All tissue samples were embedded in butyl-methyl-methacrylate (BMM) within one week.

BrdU-immunohistochemistry

Histological sections (3 µm) of the embedded sponge tissue were cut on a pyramitome (LKB 11800) using glass knives and collected on glass slides (StarFrost, Knittelglass). BMM was removed in acetone and endogenous peroxidase activity was blocked by incubating slides in methanol containing 0.3% H2O2 (20 min). Slides were washed in tris-buffered saline (TBS) and incubated in citric acid (0.2%; pH 6.0, 30 min at 85 °C). After subsequent washing in TBS, DNA was denatured in HCl (2 mol L−1, 30 min at 37 °C), pH-neutralized in sodium borate buffer (pH 8.5), and washed with TBS. Slides were incubated with mouse anti-BrdU monoclonal antibody (Nordic-MUbio MUB0200S, 1:50 in TBS with 1% BSA, 0.1% Tween 20, 60 min) then washed in TBS. Primary antibody was detected using an avidin-biotin enzyme complex (Vectastain Elite ABC Kit: Vector Laboratories, Burlingame, California, USA). Slides were incubated with biotinylated rabbit anti-mouse antibody (in TBS with 1% BSA, 0.1% Tween 20, 30 min), washed in TBS and then incubated in avidin-biotin-peroxidase complex (in TBS with 1% BSA, 0.1% Tween 20, 30 min). Peroxidase activity was visualized with DAB (DAKO; 5–10 min; positive cells have brown-stained nuclei). Sections were washed in distilled water then counterstained in hematoxylin to visualize BrdU-negative nuclei (stained dark blue), dehydrated through a graded series of ethanol, and mounted in Entellan (Merck, Kenilworth, New Jersey, USA). BrdU-labeled mouse intestinal tissue was used as a positive control and immunohistochemistry without primary antibody (on both mouse and sponge tissue) served as a negative control (Alexander et al., 2014).

Analysis of choanocyte cell kinetics

All slides were examined under a light microscope (Olympus BH-2) and photographs were taken using an Olympus DP70 camera (Olympus, Tokyo, Japan). To investigate the cell kinetics of the choanocyte population directly next to the wound (the wound tissue contained no choanocyte chambers, Fig. 2A) in early regenerative tissue, percentages of BrdU-positive choanocytes were determined at each time point: t = 0, 0.5, 1.5, 2, 6, 8 h. Three histological sections were made from each tissue sample, each approximately 100 µm apart (Fig. 1A). From each section at least 400 choanocytes were counted making a total of at least 1,200 (3 sections × 400 cells) choanocytes counted per sponge. The ‘one population model’ (Nowakowski, Lewin & Miller, 1989; De Goeij et al., 2009) was used to estimate the growth fraction (GF; percentage of choanocytes involved in proliferation), duration of the cell cycle (Tc), labeling index (LI; percentage of choanocytes in S-phase), and the duration of the S-phase (Ts) for choanocytes in early regenerative tissue. Parameters (mean ± 95% confidence intervals) were estimated from the model according to an iterated least squares fit of the data using the following specified initial conditions: (1) ft=GF×t+Ts/Tc,for

(2) t≤Tc−Ts,and

(3) ft=GF,for

(4) t⩾Tc−Ts.

Data on the cell kinetics of the choanocyte population directly adjacent to the wound in early regenerative tissue were compared to choanocyte kinetics of steady-state sponges measured by De Goeij and colleagues in 2009, as this is the only study describing detailed cell cycle parameters in H. caerulea.

Figure 2 Regenerative tissue of H. caerulea.

(A) Cross-section through BrdU- and hematoxylin-stained sponge tissue two days after wound infliction, showing regenerative tissue at the site of the wound, the location of mesohyl tracts containing cells, which were occasionally observed close to the wound, and an area away from the wound containing choanocyte chambers. Choanocyte chambers appeared 250 ± 8.9 µm from the edge of the wound tissue. (B) BrdU immunohistochemistry of a regenerative sponge labeled with BrdU for 6 h. Brdu-positive cells (brown-stained nuclei) were absent from regenerative wound tissue, and cells located in mesohyl tracts were BrdU-negative (blue-stained nuclei, white arrows). Tissue areas away from the wound that had retained their structural integrity contained BrdU-positive choanocytes (black arrows) and occasionally BrdU-positive mesohyl cells (black arrow heads). (C) Picrosirius red staining showed a higher density of collagen in regenerative wound tissue compared to areas farther from the wound. High densities of collagen could also be seen surrounding tracts in the mesohyl containing cells (white arrows). (D) Visualization of picrosirius red staining under cross polarized light revealed thin (green) and thick (orange) collagen fibers in all tissue areas.

Analysis of cell proliferation during regeneration (6 h–6 days)

In order to determine changes in cell proliferation over time, the percentages of BrdU-positive choanocytes and mesohyl cells were determined per time point: t = 0.25, 1, 2, and 6 days. Choanocytes were identified as all cells arranged in choanocyte chambers, and mesohyl cells were identified as all cells located between the choanocyte chambers and the pinacoderm. From each sponge, histological sections were analyzed from tissue samples taken directly adjacent to the wound, 1 cm from the wound, and 3 cm from the wound (Fig. 1A). Three histological sections were made from each sponge tissue sample, each approximately 100 µm apart (Fig. 1A). At least 400 choanocytes and 400 mesohyl cells were counted from each histological section making a total of at least 1,200 cells (3 sections × 400 cells) of each cell population counted per sponge in each area. Means ± SE are reported. Measuring percentages of BrdU-positive choanocytes after labeling for 6 h enables the rate of cell proliferation to be estimated, providing that the length of the cell cycle remains close to 6 h (Alexander et al., 2014). Cell proliferation during regeneration (6 h–6 days) was compared to cell proliferation from steady-state sponges kept in the same aquaria during the same experimental fieldwork period as sponges in the current study (Alexander et al., 2014).

Picrosirius red staining of collagen in regenerative tissue

Collagen content was analyzed in histological sections of sponge tissue directly adjacent to the site of the wound (Fig. 1A). BMM was removed with acetone and sections were rehydrated to distilled water, and then incubated in 0.2% phosphomolybdic acid (PMA) in water (5 min). Sections where incubated in 0.1% picrosirius red in saturated picric acid (90 min) followed by 0.01 mol L−1 HCl (2 min). Sections were dehydrated in ethanol then mounted in Entellan (Merck, Kenilworth, New Jersey, USA). Slides were viewed using a cross polarization filter in order to view thin collagen fibres (green) and thick collagen fibres (orange) (Junqueira, Montes & Sanchez, 1982). Photographs were taken of the regenerative wound tissue, and areas of tissue away from the wound (within 0.5 cm) (Figs. 2C and 2D).

Statistical analysis

Differences in choanocyte proliferation rates over time during regeneration were analyzed at each area (directly adjacent to, 1 cm, and 3 cm from the wound) using a linear model with time as a factor. Differences in choanocyte proliferation between each area were analyzed using a mixed linear model with the individual sponge as a random factor (to take into account the nested effect of area within each individual sponge). The choanocyte proliferation rate at each area after 6 days of regeneration was compared to that found in steady-state specimens of H. caerulea (data taken from Alexander et al., 2014) using a linear model. A mixed linear model was used to analyze the effect of time and area on the proliferation of mesohyl cells, and a linear model was used to determine differences in the percentages of proliferative mesohyl cells at each area during wound healing compared to steady-state H. caerulea specimens (data taken from Alexander et al., 2014). All calculations were conducted in R, using the lme4 package for mixed linear models (see Supplemental Information 3 for R scripts).

Results

All three sponges left to regenerate for six days had completely closed the wound area within this time, leaving no bare substratum exposed (Figs. 1A–1D). Six-day old tissue that filled the wound area was slightly different in color compared to fully-developed tissue. A minor depression was left at the surface of the sponge where the wound had been (Fig. 1D). Regenerative wound tissue (Fig. 2A) was void of choanocyte chambers and all cells were BrdU-negative (Fig. 2B). BrdU-positive cells and choanocyte chambers were observed to appear 250 ± 8.9 µm (mean ± SE) from the edge of the wound tissue (Fig. 2A) and were found throughout the rest of the sponge tissue (Fig. 2B). Picrosirius red staining of regenerative wound tissue revealed a high abundance of collagen fibers (a component of the extracellular matrix) compared to areas of tissue away from the wound (Fig. 2C). When viewed using cross polarization filters, picrosirius red staining showed that collagen at, and away, from the wound area consisted of both thin (green) and thick (orange) fibers (Fig. 2D). Scattered observations were made of tracts in the mesohyl close to the site of the wound, which contained BrdU-negative cells (Fig. 2B) and were surrounded by collagen (Fig. 2C).

Choanocyte cell kinetics in early regenerative tissue

The mouse intestine and sponge positive control tissues showed BrdU-positive nuclei. Negative controls (no primary antibody) and tissue samples at t = 0 (not labeled with BrdU) showed no BrdU-positive cells. The cell cycle parameters of the choanocyte population directly adjacent to the wound in early regenerative tissue were compared to choanocyte cell kinetics from H. caerulea in steady-state (data obtained from De Goeij et al., 2009) (Fig. 3 and Table 1). The number of BrdU-positive choanocytes in damaged sponges and steady-state sponges increased linearly until a maximum was reached, after which all proliferative choanocytes had been labeled and had re-entered the S-phase (Fig. 3). This maximum represents the growth fraction, i.e., the percentage of the total population of choanocytes that are proliferative, which was greatly reduced in damaged sponges (7.0 ± 2.5%) compared to steady-state sponges (46.6 ± 2.6%) (Fig. 3 and Table 1). However, the cell cycle (Tc) of choanocytes adjacent to the wound in damaged sponges was similar in length to the cell cycle of choanocytes from steady-state sponges (5.6 ± 3.4 h vs 5.9 ± 0.4 h respectively) (Fig. 3 and Table 1).

Figure 3 Choanocyte cell kinetics in early regenerative and steady-state tissue of H. caerulea.

Steady-state data were obtained from De Goeij and colleagues (2009). In both tissues, the percentage of BrdU-positive choanocytes (mean ± SE) increased over time until a maximum was reached, representing the growth fraction (GF), i.e., the percentage of choanocytes involved in proliferation. The growth fraction of choanocytes in early regenerative tissue was substantially lower than the growth fraction of choanocytes in steady-state tissue. The duration of the linear increase represents the length of the cell cycle, which was similar in regenerative and steady-state sponges. The lines are the least squares fit obtained using the conditions of the ‘one population model’ described by Nowakowski and colleagues (1989).

Table 1 Estimated cell cycle parameters (mean ± 95% confidence intervals) of choanocytes in steady-state and early regenerative tissue of H. caerulea.

Steady-state data obtained from De Goeij and colleagues (2009). The growth fraction (GF) represents the percentage of proliferative choanocytes. Tc is the length of the cell cycle, Ts is the length of the S-phase, and the labeling index (LI) represents the percentage of choanocytes in S-phase.

	Steady-state	Early regenerative	
GF (%)	46.6 ± 2.6	7.0 ± 2.5	
Tc (h)	5.9 ± 0.4	5.6 ± 3.4	
Ts (h)	0.5 ± 0.3	0.9 ± 2.5	
LI (%)	3.9 ± 1.2	1.1 ± 1.1	

Cell proliferation during regeneration (6 h–6 days)

Oscula in H. caerulea were located 1.4 ± 0.1 cm from each other. Therefore, tissue samples taken 1 cm from the wound were likely to be part of the same aquiferous module as the wound, but tissue taken 3 cm from the wound were likely to be from a separate module. Significantly less proliferative choanocytes were found closer to the wound (Fig. 4 and Table 2), i.e., less BrdU-positive choanocytes were found directly adjacent to the wound in comparison to 1 cm from the wound (mixed linear model, p < 0.001), and 1 cm compared to 3 cm from the wound (mixed linear model, p < 0.01). The percentage of proliferative choanocytes increased over time after wound infliction in all tissue areas (Fig. 4): directly adjacent to the wound (linear model, p < 0.01), 1 cm from the wound (linear model, p < 0.001), and 3 cm from the wound (linear model, p < 0.01). After six days of regeneration, there was no significant difference between choanocyte proliferation in steady-state H. caerulea tissue (17.6 ± 1.9%, data from Alexander et al., 2014) and tissue directly adjacent to the wound (12.8 ± 1.0%, linear model, p = 0.19), 1 cm from the wound (18.3 ± 0.4%, linear model, p = 0.99), and 3 cm from the wound (19.1 ± 0.5%, linear model, p = 0.94) (Fig. 4). On average, there are less BrdU-positive choanocytes directly adjacent to the wound after 6 days of regeneration compared to steady-state tissue (Fig. 4), although not statistically significant.

Figure 4 Changes in choanocyte proliferation rates over time in H. caerulea during regeneration.

There are significantly less proliferative choanocytes closer to the wound compared to 1 cm and 3 cm from the wound (mean ± SE). The percentage of proliferative choanocytes increases over time in each tissue area. The percentages of proliferative choanocytes six days after damage, in tissue located 1 cm, and 3 cm from the wound, are comparable to the percentage of proliferative choanocytes found in steady-state H. caerulea specimens (grey area represents steady-state mean ± SE; data taken from Alexander and colleagues (2014)).

Table 2 Cell proliferation during regeneration in H. caerulea.

Percentages (±SE) of proliferative choanocytes (Ch) and mesohyl cells (Me) in different tissue areas of H. caerulea during regeneration.

	Adjacent to wound	1 cm from wound	3 cm from wound	
	% Ch	% Me	% Ch	% Me	% Ch	% Me	
6h	6.9 ± 0.6	0.7 ± 0.3	8.4 ± 0.8	0.6 ± 0.8	12.5 ± 0.3	1.0 ± 0.4	
1d	7.4 ± 0.8	0.9 ± 0.6	12.1 ± 1.3	0.8 ± 0.3	15.3 ± 0.9	0.9 ± 0.3	
2d	11.9 ± 0.9	0.8 ± 0.3	13.1 ± 0.5	0.8 ± 0.4	15.4 ± 0.5	0.6 ± 0.3	
6d	12.8 ± 1.0	0.7 ± 0.2	18.3 ± 0.4	0.9 ± 0.3	19.1 ± 0.5	0.5 ± 0.2	

Between 0.5 ± 0.2% and 1.0 ± 0.4% of mesohyl cells were proliferative during regeneration. The percentage of proliferative mesohyl cells did not change over time following wound infliction (mixed linear model, p = 0.65, Table 2) and the percentages of proliferative mesohyl cells from each tissue area (directly adjacent to, 1 cm, and 3 cm from the wound) were not significantly different to each other (mixed linear model, p = 0.97) or to steady-state tissue (linear model, p = 0.10).

Discussion

Cell kinetics in early regenerative versus steady-state tissue

The choanocyte cell kinetics were altered in early regenerative tissue of H. caerulea compared to steady-state, undamaged tissue. Directly adjacent to the wound, the choanocyte growth fraction decreased 6.7 fold during initial stages of regeneration. The other cell cycle parameters remained similar in regenerative and steady-state sponges. The reduced growth fraction indicates that fewer cells participate in the proliferation and subsequent turnover of choanocytes adjacent to the wound. The eukaryotic cell cycle has several checkpoints, at which intrinsic and extrinsic signals are assessed. Under adverse environmental or physiological conditions, cells can be halted from progression through the cell cycle (Johnson & Walker, 1999; Jonas, 2014). The reduced growth fraction of choanocytes adjacent to the wound in early regenerative sponges may be due to choanocytes being halted in their progression through the cell cycle as a result of insufficient energy to progress past the G1 checkpoint. A significant amount of energy is required during cell division for the synthesis of lipids, proteins and nucleic acids (Finkel & Hwang, 2009). Studies on human cancer cell lines (Sweet & Singh, 1995; Gemin et al., 2005; Xiong et al., 2012) and cells of Drosophila larvae (Mandal et al., 2005) show that the G1 checkpoint is energetically sensitive, meaning a minimum ATP content is required to pass the checkpoint and progress through the cell cycle. Measurements of the energetics involved in regeneration for H. caerulea are required to confirm this hypothesis.

The mesohyl cells are a mixed population of cells, including archeocytes, spherulous cells, lophocytes, spongocytes, and collencytes (Simpson, 1984; Vacelet & Donadey, 1987). They may also include dedifferentiated choanocytes, endopinacocytes, and exopinacocytes, which have been shown to migrate into the mesohyl during regeneration (Diaz, 1979; Simpson, 1984). We have found that the proliferation of cells located in the mesohyl was consistently low and therefore does not appear to play a role in early stages of regeneration in H. caerulea.

Trade-off between regeneration and choanocyte proliferation

The decrease in choanocyte proliferation in regenerative sponges indicates that regeneration and choanocyte turnover (i.e., filter system renewal) are competing and negatively correlated life-history processes. Regeneration takes precedence over choanocyte turnover presumably due to the speed at which it must occur. Rapid regeneration of lost tissue in sponges is necessary to compete for limited space (Jackson & Palumbi, 1979; Turon, Tarjuelo & Uriz, 1998), prevent fouling of exposed spicules (Leys & Lauzon, 1998), and to readjust their size and shape for optimal feeding (Bell, 2002). When H. caerulea is damaged, the immediate priority is space occupation of the bare substratum, which may direct energetic resources away from choanocyte proliferation. Following wound infliction, the increase in choanocyte proliferation rates over time indicates that the energy demands of regeneration may decrease over time. The fact that choanocyte proliferation continues, although at a reduced rate, under demanding physiological conditions such as regeneration, highlights the importance of choanocyte turnover in tissue homeostasis to maintain a healthy nutrient uptake system (De Goeij et al., 2009; Alexander et al., 2014). Rapid choanocyte turnover has been documented in multiple sponge species from tropical reef, mangrove, and temperate ecosystems (Alexander et al., 2014). However, sponges vary widely in their ability to regenerate and recover from injury depending on their morphology, structural complexity, susceptibility to damage, and chemical defenses (Wulff, 2006a; Walters & Pawlik, 2005; Wulff, 2010). Therefore, the relationship between regeneration and choanocyte renewal may vary depending on the species.

After six days of regeneration, choanocyte proliferation rates in regenerative sponges were comparable to that of steady-state, undamaged sponges (17.6 ± 1.9%) measured in the same running-seawater aquaria during the same field work campaign as the sponges used in the current study (Alexander et al., 2014). However, these proliferation rates are substantially lower than rates measured from sponges kept in the same aquaria in 2009 (46.6 ± 2.6%; De Goeij et al., 2009). Our results corroborate the suggestion that the lower proliferation rates measured in steady-state sponges by Alexander and colleagues (2014) result from a reduced growth fraction rather than a change in the length of the cell cycle. We hypothesize that suboptimal nutritional conditions are the cause of these differences (Alexander et al., 2014) and this relationship needs to be analyzed further. Nutrition plays an important role in wound healing in humans (reviewed by Reynolds, 2001), and the allocation of energy to limb regeneration has been found to depend on levels of available food in the asteroid Luidia clathrata (Lawrence et al., 1986; Lawrence & Ellwood, 1991). Low food availability can exaggerate energy trade-offs in insects (reviewed by Zera & Harshman, 2001), and therefore under optimal nutritional conditions, the trade-off between regeneration and choanocyte proliferation in H. caerulea may not be as evident.

Regeneration is a common phenomenon occurring in many metazoan tissues; however, the extent to which tissues can regenerate to a fully functional state and the cellular mechanisms involved in regeneration vary widely (reviewed by Tanaka & Reddien, 2011). In many cases, the proliferation of stem cells increases during regeneration. Highly proliferative tissues often maintain a pool of quiescent stem cells that divide in response to injury e.g., in mammalian intestinal stem cells (Buczacki et al., 2013) and hair follicle stem cells (Li & Clevers, 2010), as well as in head regeneration in Hydra resulting from mid-gastric amputation (Govindasamy, Murthy & Ghanekar, 2014). Regeneration can also occur through the rearrangement of pre-existing tissue in the absence of cell proliferation, e.g., in head regeneration following decapitation in Hydra (Holstein, Hobmayer & David, 1991; Bosch, 2007). Regeneration in sponges has generally been shown to be a result of remodeling and reorganization of pre-exisiting tissue involving both collagen production and the migration of cells to the wound in mesohyl tracts (e.g., Harrison, 1972; Boury-Esnault, 1976; Jackson & Palumbi, 1979; Simpson, 1984; Smith & Hildemann, 1986; Louden et al., 2007). We have shown that regeneration in H. caerulea does not involve an increase in cell proliferation. Although the cellular mechanisms responsible for the production of early regenerative tissue in H. caerulea remain unknown, collagen production and mesohyl tracts are likely to play a role.

Organismal integration during regeneration

The degree of integration between aquiferous modules during regeneration in sponges is unclear, with some evidence suggesting that resources required for regeneration are derived from tissue in the immediate vicinity of the wound (Boury-Esnault, 1976; Wulff, 1991). Others suggest that resources may be contributed from a more extensive distance from the wound (Boury-Esnault & Doumenc, 1978; Wulff, 2006b; Henry & Hart, 2005), as occurs in other modular organisms such as corals (Oren, Rinkevich & Loya, 1997; Oren et al., 2001). Integration between modules during regeneration in sponges may be dependent on the size of the wound. In the coral Favia favus, the regeneration of small wounds (<1 cm2) demands only localized resources while regeneration of larger wounds involves an integrated response of the organism (Oren et al., 2001). We have shown that during the regeneration of small wounds in H. caerulea, the proliferation of choanocytes located farther from the wound were less affected by the process of regeneration. This suggests that the regeneration of small wounds demands resources mainly from tissue in their immediate vicinity. Since choanocyte proliferation rates are reduced during initial stages of regeneration, even at a distance of 3 cm from the wound, we conclude that some degree of integration between modules occurs. In order to measure the full extent of organismal integration, the effect of regeneration on choanocyte proliferation should be determined using larger specimens of H. caerulea, which we have observed growing in situ up to meters in size, and using larger wounds. The organization of sponge tissue into repeated functional units (modules) allows multioscular sponges to function and survive if separated into smaller pieces due to fragmentation (Ereskovskii, 2003). However, the ability to provide an integrated physiological response during periods of stress is an important survival strategy for modular organisms (Oren et al., 2001).

Conclusions

This study demonstrates the inhibitory effect of regeneration on cell proliferation in H. caerulea, and the gradual return over time to proliferation rates found in steady-state, undamaged tissue. We provide evidence for a trade-off between choanocyte proliferation and regeneration. Furthermore, these results demonstrate that the impact of regeneration on cell proliferation decreases farther from the wound, indicating that the wound demands more resources from tissue in its immediate vicinity. An understanding of the relationship between regeneration and choanocyte renewal, two life-history traits essential to the survival of sponges, allows us to gain insight into the physiological functioning responsible for the ecological success of one of the first extant multicellular organisms on Earth.

Supplemental Information

Supplemental Information 1 Raw data choanocyte cell kinetics during regenerative and steady-state conditions in H. caerulea

Click here for additional data file.

Supplemental Information 2 Raw data of choanocyte and mesohyl cell proliferation during regeneration (6 h–6 days) and during steady-state conditions

Click here for additional data file.

Supplemental Information 3 R script showing models used for statistical analyses

Click here for additional data file.

We thank Fabiènne Doveren and Kevin Liebrand for their assistance in the field and lab; staff at the CARMABI institute for their hospitality during field work; Henk van Veen, Wikky Tigchelaar and the staff of the electron microscopy lab at the Academic Medical Center (AMC) Amsterdam for their support in histology; the Earth Surface Sciences Department at the Institute for Biodiversity and Ecosystem Dynamics (ESS-IBED), University of Amsterdam for access to their microscopy facilities; Emiel van Loon for advice on statistical analyses and Henk Kieft for help with BMM embedding.

Additional Information and Declarations

Competing Interests

Author Contributions

Field Study Permissions

Brittany E. Alexander, Ronald Osinga and Jasper M. de Goeij are employees of Porifarma B.V., Poelbos, The Netherlands.

Brittany E. Alexander conceived and designed the experiments, performed the experiments, analyzed the data, wrote the paper, prepared figures and/or tables, reviewed drafts of the paper.

Michelle Achlatis performed the experiments, analyzed the data, reviewed drafts of the paper.

Ronald Osinga and Bert Schutte conceived and designed the experiments, contributed reagents/materials/analysis tools, reviewed drafts of the paper.

Harm G. van der Geest conceived and designed the experiments, reviewed drafts of the paper.

Jack P.M. Cleutjens conceived and designed the experiments, performed the experiments, contributed reagents/materials/analysis tools, reviewed drafts of the paper.

Jasper M. de Goeij conceived and designed the experiments, performed the experiments, analyzed the data, contributed reagents/materials/analysis tools, wrote the paper, reviewed drafts of the paper.

The following information was supplied relating to field study approvals (i.e., approving body and any reference numbers):

Fieldwork on Curaçao was performed under the research permit (#2012/48584) issued by the Curaçaoan Ministry of Health, Environment and Nature (GMN) to the CARMABI foundation.

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
