# Peer review of "Cell kinetics during regeneration in the sponge Halisarca caerulea: how local is the response to tissue damage?"

_PeerJ, doi:10.7717/peerj.820_

## Round 0.1 · original submission · Major Revisions

· Academic Editor

Major Revisions

Discussion section on regeneration of sponge wound region, specifically choanocytes damaged can be extensively added for the cell cycles and apoptosis compared with the vertebrates, as compared to the undamaged tissues.

Reviewer 1 ·

Basic reporting

General Comments
The authors have conducted a careful study of wound regeneration in an encrusting coral reef sponge. They report that during the process of regeneration the number of proliferative choanocytes adjacent to the wound is reduced when compared to undamaged tissue. These studies appear to have been carefully conducted.
The authors hypothesize that reduction of choanocytes is likely due to “Insufficient energy resources to progress through the cell cycle”. The bulk of the paper argues for support of this hypothesis. However, no actual data is provided to support this observation beyond the observation of reduced choanocyte proliferation. That is, no experiments related to energetics were conducted.
The available data provides a nice correlation. However, there is not data demonstrating a cause-and-effect.

Specific Comments
The introduction is unnecessarily long in the description of sponge biology.
Please include data on the light/dark cycle of the exposure. Was the seawater filtered? What depth were the sponges covered with water?
The following is not going to be common to the typical reader. ‘All calculations were conducted in R, using the lme4 package for mixed 225 linear models (R Development Core Team 2013). Please provide further details on methods of analysis.

Line 230-231. What accounts for the color difference compared to fully developed tissue?

The discussion, while informative, overstates the case for the autho’rs energetic argument in the absence of any data. While it is possible, even likely, the authors may be correct in their energetics hypothesis…they have not designed or conducted experiments that would test this hypothesis.

Experimental design

Additional detail on the system for holding the sponges would be useful.

Additional experiments related to measurement of energy flow through the sponges would add quite a bit of support to the author's hypothesis.

Validity of the findings

no comment

Additional comments

While I believe you have meet the qualification for publishing in this specific journal, I do not think you have provided adequate data to support your energetics hypothesis. As such, you should edit such that you are not overstating your conclusions when based on the available data.

Reviewer 2 ·

Basic reporting

This manuscript presents the results of the investigation of spatial relationship between regeneration and cell proliferation in the encrusting reef demosponge Halisarca caerulea over a six-day period following experimental wound infliction as well as speed rates of body part regeneration. I think this manuscript is worth publishing in PeerJ as a Basic Reporting. The manuscript is well-written: in general it is clear and the quality of the English is generally good. It is sufficiently illustrated. This paper includes new observations and thereby makes a contribution to our knowledge of sponge cell proliferation rate and of body part regeneration. All sections of the manuscript satisfy expected criteria. This paper should be of interest to both sponge biologists and the specialist in cell kinetics and animal regeneration in natural conditions. Nevertheless I have criticisms in this paper and I recommend that it should be accepted for publication, only after amendment as itemized below.

General comments
My principal criticism for this MS concerns to authors reasoning with respect to the mechanisms of regeneration (morphallaxis) during regeneration of sponges in general and Halisarca caerulea in particular: Introduction – lines 61-71 and Discussion, lines 286-310.
Usually in the works related to the mechanisms of regeneration, primarily focuses on the tissue and cellular aspects of this process, as well as the peculiarities of cell’s and epithelia movements. However, in the MS of Alexander et al. there are no description of the cellular composition of the regenerate, do not mentioned presence or absence of a blastema, no any mention of dedifferentiation, transdifferentiation, morphogenesis etc. It was not show any features of cell movements, it is not known which cells participate at different stages of regeneration. Not description of different stages of Halisarca caerulea regeneration. The principal focus in this MS is on the cell proliferation dynamics during regeneration. That is why, I suggest either completely remove the reasoning concerning of regeneration mechanisms, or include in the article the results of electron microscopic studies of regeneration: cellular dynamics, morphogenesis, etc. It seems to me that the article has nothing to lose if the reasoning, given above (regeneration mechanisms), will be deleted. Moreover, the article will be more seamless. The results presented here on cell proliferation are very good and are of great interest to specialists.

Experimental design

see attachment review

Validity of the findings

see attachment review

Additional comments

see attachment file

Annotated reviews are not available for download in order to protect the identity of reviewers who chose to remain anonymous.

---

## Round 0.2 · accepted · Accept

· Academic Editor

Accept

Your have appropriately revised the manuscript with the comments raised.